# Phenolics from Defatted Black Cumin Seeds (*Nigella sativa* L.): Ultrasound-Assisted Extraction Optimization, Comparison, and Antioxidant Activity

**DOI:** 10.3390/biom12091311

**Published:** 2022-09-16

**Authors:** Abdelkrim Gueffai, Diego J. Gonzalez-Serrano, Marios C. Christodoulou, Jose C. Orellana-Palacios, Maria Lopez S. Ortega, Aoumria Ouldmoumna, Fatima Zohra Kiari, Georgia D. Ioannou, Constantina P. Kapnissi-Christodoulou, Andres Moreno, Milad Hadidi

**Affiliations:** 1Department of Process Engineering, Faculty of sciences and Technology, University of Mustapha Stambouli Mascara, Mascara 29000, Algeria; 2Department of Organic Chemistry, Faculty of Chemical Sciences and Technologies, University of Castilla-La Mancha, 13071 Ciudad Real, Spain; 3Department of Chemistry, University of Cyprus, Nicosia 1678, Cyprus

**Keywords:** phenolic compounds, antioxidant, optimization, ultrasonic, extraction techniques, black cumin

## Abstract

An ultrasound-assisted method was used for the extraction of phenolics from defatted black cumin seeds (*Nigella sativa* L.), and the effects of several extraction factors on the total phenolic content and DPPH radical scavenging activity were investigated. To improve the extraction efficiency of phenolics from black cumin seed by ultrasonic-assisted extraction, the optimal extraction conditions were determined as follows: ethanol concentration of 59.1%, extraction temperature of 44.6 °C and extraction time of 32.5 min. Under these conditions, the total phenolic content and DPPH radical scavenging activity increased by about 70% and 38%, respectively, compared with conventional extraction. Furthermore, a complementary quantitative analysis of individual phenolic compounds was carried out using the HPLC-UV technique. The phenolic composition revealed high amounts of epicatechin (1.88–2.37 mg/g) and rutin (0.96–1.21 mg/g) in the black cumin seed extracts. Ultrasonic-assisted extraction can be a useful extraction method for the recovery of polyphenols from defatted black cumin seeds.

## 1. Introduction

Black cumin (*Nigella sativa* L.), an herbaceous plant member of the Ranunculaceae family, is one of the most popular medicinal herbs in the north of Africa, with wild populations distributed in Asia, Southern Europe, Northern Africa, and the Mediterranean area. Black cumin seeds have a long history in pharmacotherapy as a medicinal herb used in traditional remedies to treat a broad variety of diseases, including diabetes, different airway disorders, paralysis, digestive tract issues, inflammation, and hypertension [1]. These medicinal properties of black cumin seeds are directly connected with their high content of phenolic compounds, which act as antioxidants and display an extremely anti-inflammatory capacity [2]. Oilseeds are important sources of lipid, commonly processed for the extraction of edible oils, producing large quantities of bioactive compound-rich meal [3]. Thus, defatted black cumin seeds (black cumin seed meal) could be an interesting source of this type of compound.

One of the key indicators for several progressive pathological illnesses, such as neurological problems and endocrine sickness, is oxidative stress, which is characterized by a rise in free radical levels [4]. The therapeutic efficacy of medicinal herbs acting as scavengers of free radicals has recently gained more attention. Studies conducted both in vivo and in vitro on black cumin seed extracts have shown that they have strong antioxidant properties [5]. Similarly, an indicator of the remarkable bioactive capacity of black cumin seeds could be found in their traditional consumption for the treatment of a diverse range of illnesses [6]. Thus, black cumin extracts may be a possible source of innovative antioxidant and anti-inflammatory phenolic compounds, safer than current medications, which frequently, after prolonged use, cause serious negative consequences such as gastric ulcers, bone marrow depression, and water and salt retention [7].

Recently, studies focused on identifying different fruit, vegetables, and plants as new sources of bioactive components have increased considerably since these natural products have a broad variety of health benefits for humans [8,9]. Such components are usually obtained through a lengthy and expensive process that includes extraction, isolation, and identification, frequently leading to the heat breakdown of certain bioactive components [10]. Natural phenolic compounds have tremendous potential for prevention, preservation, and therapy, which has led to a surge in interest regarding their appropriate extraction method [11]. Techniques used for both conventional and non-conventional bioactive component extraction have been distinguished. Traditional or conventional extraction methods present some important drawbacks, including higher extraction time and more solvent use [12]; non-conventional techniques, such as microwave-assisted or ultrasound-assisted extraction, have proved to be not only more sustainable but also more effective alternatives [13,14].

Ultrasound-assisted extraction is one of the most safe, rapid, and efficient non-conventional extraction methods. In many cases, ultrasound irradiation minimizes the destruction of thermosensitive chemicals, also using less solvent, taking less time, and producing greater extraction yields [15]. Ultrasound treatment implies the breakage of cell walls through the bursting of bubbles created by sonic cavitation, improving mass transference of the cellular content to the solvent and thus increasing the extraction yield of bioactive compounds [13]. Ultrasound-assisted extraction is also frequently used in the food industry since it is a safe and green way to speed up chemical processes [16].

Response Surface Methodology (RSM) is commonly used to optimize extraction processes. Although the traditional approach requires excessive time and yields insufficient information, RSM is a quick alternative that offers sufficient data for a wide variety of independent variables, also considering the interaction between these factors [17,18]. In this regard, RSM is defined as a collection of statistical and mathematical tools used to model and analyze problems whose responses are influenced by different variables. The primary benefit of this tool is the use of only a small number of statistically significant experiments to evaluate numerous factors and their interactions, saving time and providing a large amount of data [19,20].

To the best of our knowledge, the current study represents the first attempt to optimize the ultrasound-assisted extraction of phenolic compounds from defatted black cumin seeds, seeking to maximize total phenolic compounds (TPC) and antioxidant activity (DPPH method). For this purpose, RSM was selected to conduct the optimization process, employing the following three-level independent factors: temperature (25, 50, and 75 °C), time (15, 30, and 45 min) and ethanol concentration (30, 60, and 90%). The results achieved were also compared with those obtained by the conventional method. Finally, the extracts recovered in both optimal and conventional conditions were analyzed by the High-Performance Liquid Chromatography-Diode Array Detector (HPLC-DAD) to identify and quantify the different phenolic compounds.

## 2. Materials and Methods

### 2.1. Chemicals and Reagents

All chemicals and reagents used in this study were analytical grade. The chemicals such as 2,2′-diphenyl-1-picrylhydrazyl radical (DPPH•), absolute ethanol (99.9%, *v*/*v*) and sodium carbonate (Na_2_CO_3_) were purchased from Merck (Darmstadt, Germany). LC-MS grade water, ethanol and acetonitrile were obtained from Sigma–Aldrich (Steinheim, Germany), while trifluoroacetic acid (TFA) bought from Merck (Darmstadt, Germany). Catechin, myricetin, and quercitrin were provided by HWI ANALYTIK GMBH, while quercetin, epicatechin, sinapic acid, caffeic acid and gallic acid were acquired from Sigma–Aldrich (St. Louis, MO, USA). Rutin was purchased from PhytoLab GmbH & Co (Vestenbergsgreuth, Germany). Black cumin (*Nigella sativa* L.) seeds were obtained from a local market in Mascara City (Mascara, Algeria) in March 2022.

### 2.2. Extraction of Phenolic Compounds from Black Cumin

#### 2.2.1. Ultrasound-Assisted Extraction

Seeds were ground using a Pulverisette 14 mill (FRITSCH, Idar-Oberstein, Germany) to obtain a fine powder. Then, the obtained power was defatted with hexane at room temperature (ratio of 1:20 *w*/*v*). The powder was kept in a clean, dried, well-sealed amber glass container to protect it from sunlight and stored in the refrigerator at 4 °C for further use.

A total of 5 g of the sample was combined with 200 mL of ethanol/water solution at different concentrations (30, 60, and 90%, *v*/*v*) to extract the phenolic components from the black cumin seeds. The samples were extracted using an ultrasonic bath (model UP200S, Hielscher, Berlin, Germany) that was partially filled with distilled water (about 3.0 dm^3^). The extraction procedures were carried out using three different ultrasonic temperatures of 25, 50, and 75 °C, for 15, 30, and 45 min, with a continuous power of 110 W and a frequency of 40 kHz. Then, the extracts were filtered with Whatman No. 1 filter paper at room temperature. The filtered solution was recovered and evaporated using a rotary evaporator for 30 min at 40 °C. The remaining samples were freeze-dried at −80 °C for 24 h and kept at 4 °C for further experiments. The sample redissolved in the solvent after evaporation to obtain 2 mg/mL of each extract.

#### 2.2.2. Conventional Extraction

The conventional extraction method followed the procedure described by Bouaoudia-Madi et al. [21] with some minor modifications. Briefly, 5 g of sample was placed in a conical flask and mixed with 200 mL of 50% (*v*/*v*) ethanol solution. Then, the mixture was stirred for 2 h at 50 °C, filtered employing Whatman No. 1 filter paper and finally stored at 4 °C for further analysis.

### 2.3. Determination of DPPH Radical Scavenging Activity

DPPH radical scavenging activity was measured according to the method of Li, Du, Jin, and Du [10]. Briefly, 2 mL of 0.25 mmol/L DPPH• methanolic solution was mixed with 1 mL of black cumin seed extract, vortexed, and left at room temperature (25 °C) in a dark place for 30 min. The control was made without extracts, and both the sample mixtures and the control were measured for absorbance at 517 nm (V-750 spectrophotometer, Jasco, Tsukuba, Japan). The following equation was used to calculate the percentage of DPPH• that black cumin seed extracts were able to scavenge.
$$DPPH\;radical\;scavening\;activity\;\left(\%\right)=\frac{\left(A_{control}-A_{sample}\right)}{A_{control}}\;\times \;100$$
where *A_control_* is the absorbance of DPPH• methanolic solution and *A_sample_* is the absorbance of DPPH• solution mixed with the sample extract.

### 2.4. Determination of Total Phenolic Content (TPC)

The TPC of the samples was assessed using the Folin–Ciocalteu assay based on the Zakaria et al. method [19] with some modifications. A total of 1 mL of sample containing 1.0 mg/mL of a standard gallic acid solution was mixed with 0.5 mL of the Folin–Ciocalteu reagent for every extract, and the mixture was left at room temperature for 3 min. The Na_2_CO_3_ solution was prepared at a concentration of 7.5% (*w*/*v*) then heated for 1 min at 95 °C and cooled at room temperature. Next, 1.0 mL of 7.5% Na_2_CO_3_ was added to the sample or gallic acid solution. After 1 h of incubation at room temperature in a dark place, the absorbance of the sample was determined at 760 nm using a UV-Vis spectrometer (V-750 spectrophotometer, Jasco, Tsukuba, Japan). Gallic Acid (mg) was used as a standard to plot the calibration curve. TPC of the extracts was expressed as mg gallic acid equivalents (GAE) per gram of sample in dry weight (mg/g).

### 2.5. Determination of Phenolic Profile

#### 2.5.1. Method Validation

The current analytical method was validated according to linearity, limit of detection, limit of quantification and precision in terms of reproducibility (intraday validation) and repeatability (interday validation). For intraday assay, six successive HPLC analyses of standard solutions were performed on the same day, while in the case of interday validation, the analysis was conducted on five consecutive days. At least five different concentrations were prepared for each analyte and the construction of calibration curves. The linear ranges for each analyte are represented in Table 1. The limits of detection (LOD) and limit of quantification (LOQ), used to assess the sensitivity of the method, were calculated based on the standard deviation of the response (σ) and the slope of the corresponding calibration curve [22]. The detection limits were expressed as:(1)LOD = 3.3 σ/Slope(2)LOQ = 10 σ/Slope

#### 2.5.2. HPLC-UV Analysis

For the chromatographic analysis, 8 mg of the freeze-dried samples were dissolved in 1 mL of ethanol (HPLC-MS grade). The final solutions were filtered through 0.45 μm pore size PTFE syringe filters (LLG-Syringe filters SPHEROS, Mechenheim, Germany) and analyzed via the HPLC-UV system. HPLC-UV analysis was performed according to a previously reported method [23]. The polyphenolic existence of nine compounds was performed using an HPLC system from Shimadzu (Kyoto, Japan), equipped with a pump (LC-10AD), a PDA detector (SPD-M20A), an autosampler (SIL-20AHT) and a thermostat column compartment (CTO-10ASVP). The PDA detector was set at 280 nm for the determination of all analytes. For the chromatographic separation, a Venusil XBP C18 column (150 × 4.6 mm, 5 m) and pre-column composed of the same material were used. Mobile phase A consisted of Milli Q water while mobile phase B was ACN with 0.02% TFA. The method operated with a flow rate of 1.0 mL/min, while the injections were 20 μL. The initial conditions of the gradient elution were 80% A for the first five min, followed by a stepwise decrease to 60% A until 8 min, after which there was a further decrease to 50% A until 12 min, and finally an increase to 60% until 17 min. The ratio of mobile phases was reset to the original composition at 21 min, where it remained constant until 25 min.

### 2.6. Response Surface Methodology (RSM)

The Box–Behnken design (BBD)-based response surface technique was applied to optimize the UAE parameters for the extraction of bioactive components from black cumin seeds. A total of 16 research experiments were carried out in the BBD. The ethanol concentration (X_1_) (30, 60, and 90%), sonication temperature (X_2_) (25, 50, and 75 °C), and sonication time (X_3_) (15, 30, and 45 min) were the independent variables. The responses included the TPC and DPPH radical scavenging capacity. Table 1 displays the whole design matrix of UAE extraction variables and responses together with their corresponding levels and coded components. A second-order polynomial model was used to fit the responses of triplicate measurements of TPC and DPPH radical scavenging activity as follows:Y = β_0_ + β_1_X_1_ + β_2_X_2_ + β_3_X_3_ + β_11_X_1_ + β_22_X_2_ + β_33_X_3_ + β_12_X_1_X_2_ + β_13_X_1_X_2_ + β_23_X_2_X_3_
where X_1_, X_2_, and X_3_ represent the independent variables ethanol concentration, sonication temperature, and sonication time, respectively, and Y represents the expected responses (TPC and DPPH radical scavenging activity) from the black cumin extracts. A constant is represented by β_0_; the linear regression coefficients are β_1_, β_2_, and β_3_; the interaction terms are β_12_, β_13_, and β_23_; and the quadratic coefficients are β_11_, β_22_, and β_33_. To visualize the relationships between the values of each independent variable and the responses, three-dimensional (3D) charts were made by using the polynomial equations.

## 3. Results and Discussion

### 3.1. Model Fitting and Statistical Analysis

The total efficiency of the UAE is determined by different experimental factors, including extraction temperature, time, and ethanol concentration. The results of 16 experimental runs using the Box–Behnken design are presented in Table 1, which contains the measured values for both responses (Y_1_ and Y_2_) for each trial. It is well recognized that certain variables, such as the sonication temperature, time, and ethanol concentration, have a substantial influence on the recovery of total phenolic content and antioxidant activity from plant materials. The TPC and DPPH radical scavenging activity values for black cumin seed were 19.2–35.6 mg GAE/g and 35–70.5%, respectively. This strongly suggests that the extraction conditions have a significant impact on the yield of these parameters. These results also indicate a substantial sensitivity of the recovery of phenolic antioxidants to the extraction conditions, thus highlighting the necessity to optimize UAE parameters to obtain maximal TPC and DPPH radical scavenging activity from black cumin seed [24].

Table 2 presents the ANOVA results, model adequacy, and regression coefficients. Additionally, various statistical indicators were used to assess the model adequacy, including the coefficient of correlation (R^2^), adj-R^2^, and the coefficient of variation (CV). The coefficient of correlation (R^2^) values of TPC and DPPH radical scavenging activity were 0.992 and 0.972, respectively. R^2^ value was used to judge the model adequacy. The developed models that included both TPC and DPPH radical scavenging activity had a *p*-value of 0.0001, which means that they were significant. Diagnostic plots (Figure 1) are employed to investigate the model satisfactoriness and the relationship between predicted and experimental values. It is evident from Figure 1 that the data points lie very closely to the straight line with a high degree of similarity. A high correlation between the predicted and experimental data reflects the applicability and accuracy of RSM for optimization of the extraction process. Furthermore, the adj-R^2^ values (0.9811–0.9320) show the model to be significant. On the other hand, low values of CV (3.2–7.16%) indicate that the actual values are highly precise and reliable [20]. In most cases, it is necessary to verify that the fitted model gives a sufficient approach to the real system. The findings demonstrated that the models employed in this research were fitted, and the results of the experiment were reliable and exact for the prediction and optimization of UAE parameters to achieve a higher TPC and DPPH radical scavenging activity.

### 3.2. Influence of Process Variables on the TPC

Phenolic compounds can function as antioxidants due to their capacity to chelate metal ions and donate hydrogen atoms or electrons to stabilize free radicals. This prevents food products, particularly oils and fatty acids, from oxidizing [19]. The values of the experimental data along with the evaluation of the impact of independent factors on the TPC of black cumin extract are shown in Table 1. The analysis of variance (ANOVA) findings indicated that the model was adequate for evaluating TPC, with a coefficient of determination (R^2^) of 0.9924 (Table 2) [25]. The experimentally determined levels of TPC in black cumin seed varied from 19.2 mg GAE/g to 35.6 mg GAE/g. The circumstances in the UAE of 60% (ethanol concentration), 50 °C (temperature), and 30 min (time) resulted in the highest TPC (35.6 mg GAE/g). The TPC found in black cumin seed was higher than those observed in the following solanum species: *Solanum ferrugineum* (31.41 mg GAE/g), *S. melongena* (16.97 mg GAE/g), and *S. betaceum* (24.74 mg GAE/g) [26]. However, it was lower than the Gandhi et al. [27]-reported value of 280 mg GAE/g in the *S. torvum Swartz* extract. This may be caused by regional variances, the extraction method and conditions used. The following second-order polynomial equation was created to examine the impact of the independent variable on the extraction of TPC (Equation (1)).
Y_TPC_ = +35.25 − 1.14X_1_ − 2.29 X_2_ + 1.10X_3_ − 1.72 X_2_X_3_ − 6.78X_1_^2^ − 6.02X_2_^2^ − 5.85X_3_^2^(1)

According to Table 2, the ethanol concentration (X_1_), extraction temperature (X_2_), and extraction time (X_3_) all had a significant effects on the total phenolic content of black cumin seed (*p* < 0.05, *p* < 0.01, and *p* < 0.01, respectively). Moreover, the impacts of the quadratic functions (X_1_^2^, X_2_^2^ and X_3_^2^) and the interaction between extraction time and temperature (X_2_X_3_) had a significant (*p* < 0.05) impact on the UAE of TPC from black cumin extracts. Furthermore, the extraction yield of TPC from black cumin seed was unaffected by the interactions between the concentration of ethanol and time and the concentration of ethanol and temperature (*p* > 0.05).

As shown in Figure 2a, the TPC improves as the ethanol concentration increases from 30 to 60%. On the other hand, when the ethanol concentration increases to 90%, the TPC decreases regularly to less than 27 mg/g. According to Gullon et al. [28], these effects are a result of the polarity of the solvent having an impact on the extraction of phenolics. These authors also showed that the TPC of the extract from *Eucalyptus globulus* leaves was affected by a similar trend. In addition, when using ultrasound-assisted extraction, ethanol concentration had a similar dual impact on the TPC isolated from Tunisian *Zizyphus lotus* fruit [29]. Figure 2a shows that the sonication time has the most impact on the recovery of TPC. The extraction time was adjusted from 15 to 45 min to demonstrate how it affected TPC efficiency. The TPC increased when the extraction time was raised to 30 min. However, longer times implied a decrease in TPC. In a similar study, Irakli et al. [30] observed an increase in TPC in the first 30 min of extraction by UAE and then a sharp decrease in TPC. The sonication time is divided into two main stages when secondary metabolites are extracted by UAE. Most of the retrieved metabolites occur during the initial “washing stage” (20–30 min). Then comes the “slow extraction phase” which lasts for 60 to 100 min, when metabolites are transported by a diffusion process from the matrix to the solvent [31]. This two-stage division of the extraction could explain the results obtained, considering that the diffusion process may adversely affect the extraction yield when most of the phenolic compounds are already extracted [32]. Additionally, one of the key elements in the UAE is the sonication temperature. In general, an elevation in this variable is correlated with increases in phenolic compound extraction yields. This is due to the breaking down of matrix bonds, increasing mass transfer, chemical solubility, and solvent diffusion velocity while decreasing solvent viscosity and tension [33]. The sonication temperature was varied from 25–75 °C. Figure 2a showed that the yield of the extraction of phenolic compounds from black cumin seeds increased markedly over the temperature range of 40–50 °C. The highest TPC yield (35.6 GAE/g dm) was obtained at 50 °C, this temperature being picked as the optimum value. As a result of the solvent molecules moving more quickly and easily at higher temperatures, the extraction of phenolic compounds increases as the extraction temperature is raised [34]. However, heat-sensitive components will be damaged if the extraction temperature is too high [35]. This could explain why TPC content dropped at temperatures higher than 50 °C.

### 3.3. Influence of Process Variables on the DPPH Radical Scavenging Activity

According to Madhujith and Shahidi [36], phenolic compounds produced from plant sources are widely used and significant due to their ability to serve as reducing agents, free radical scavengers, and metal ion chelators to reduce and control oxidative stress. The most common technique for the evaluation of antioxidant activity in plants is the DPPH radical scavenging activity assay. For this reason, DPPH radical scavenging activity was used to examine the antioxidant potential of defatted black cumin seeds. As it can be seen in Table 1, black cumin extracts report an antioxidant activity of between 35 and 70.5%. The highest percentage of DPPH radical scavenging activity (70.5%) was presented in extract 4 (Table 1), in the conditions of 60% ethanol concentration, 50 °C temperature and 30 min of sonication duration. In addition, regression analysis using ANOVA revealed that the chosen model was very significant since the *p*-value was <0.0001 (Table 2). Furthermore, model coefficients illustrated that all the linear factors had a substantial impact (*p* < 0.01 and *p* < 0.001) on the result. The following polynomial equation (Equation (2)) demonstrates the relationship between process factors and the antioxidant activity of the extracted compounds:Y_DPPH_ = +69.15 + 12.75X_1_ + 11.93X_2_ + 34.53X_3_ + 9.67X_1_X_3_ + 4.03X_2_X_3_ + 62.04X_1_^2^ + 46.66 X_2_^2^ + 30.82 X_3_^2^(2)
where Y_DPPH_ stands for the black cumin seed extract ability to scavenge DPPH radicals, and X_1_, X_2_, and X_3_ stands for the ethanol concentration (%), temperature (°C), and time (min), respectively.

Three-dimensional response surface plots were also created to analyze the relationships between the experimental data of the investigated variables and the DPPH radical scavenging activity. The results demonstrated that the extraction process variables significantly affected the DPPH radical scavenging activity (Figure 3b). Even though the interaction between ethanol concentration and temperature (X_1_X_2_) had no effect on DPPH radical scavenging activity, it was proven that the interactions between ethanol concentration and time (X_1_X_3_) and between temperature and time (X_2_X_3_) had a significant impact (Figure 3b). As it can be seen in Figure 2b, the antioxidant activity content of the resultant extracts increased significantly in three cases: when the ethanol concentration approaches 60%, when the extraction period is extended from 15 to 30 min, and when the temperature is raised from 30 to 50 °C. According to Fadimu et al. [37], the UAE process promotes the dissolution of phenolic compounds as the ethanol concentration rises. In combinations of water and ethanol, water works as a swelling agent, whereas ethanol breaks down the links between solutes and the cellular matrix [38]. As a result, increasing the ethanol concentration would lead to the extraction of more bioactive compounds with high antioxidant activity. However, even though a 60% ethanol concentration was the optimal value, the DPPH radical scavenging activity was quickly lowered to the minimal value of 35% as the ethanol concentration increased to 90%. This phenomenon may be caused by the poor solubility of phenolic compounds in ethanol, since it appears that a larger concentration of this solvent is not appropriate for extracting more potent antioxidants due to their polarity. The same results were also observed in a study by Prasad et al. [39] as ethanol concentrations exceeded 68%. Additionally, as previously studies reported by Sahin and Samli and Tomsik et al. [38,40], when temperature rises over 60 °C, DPPH radical scavenging activity is decreasing. This reduction is a possible outcome by the breakdown of some heat-sensitive phenolic compounds in the matrix. Furthermore, according to the sonication time, TPC results were in align with DPPH radical scavenging activity, with a maximum value at 30 min, and subsequently decreased at a higher time (45 min). These mild conditions are expected since higher sonication temperatures and longer sonication times than optimal levels may decompose sensitive phenolic compounds [11]. Additionally, DPPH radical scavenging activity assay had results that were remarkably comparable to those found in previous research studies [11]. Surprisingly, the DPPH radical scavenging activity value of black cumin was significantly closer to that of date palm spikelets extract (87.2%). It was obtained at optimal UAE conditions of 40.8 °C, 21.6 min, and 50.0% ethanol–water concentration. In another study, conducted by Bouafia et al. [41], the following parameters of 38.9 min, 54.7 °C, and a ratio of 45.2 mL/0.5 g resulted in the highest antioxidant capacity for UAE model (42.17 mg AAE/g dm).

### 3.4. Phenolic Compound Profile

Regarding our results, the obtained regression equations revealed a linear relationship between the peak area of each polyphenolic compound and their concentration, with corresponding correlation coefficients higher than 0.995. In the meantime, RSD% values of intra and interday assays, were, respectively, in the ranges of 0.32–1.04 and 0.92–2.68. The limits of detection and quantification were in the range of 0.1–0.9 and 0.3–2.8 µg/mL, respectively, as is illustrated in Table 3. The current HPLC-UV analysis examined the possible existence of nine major polyphenols in the UAE and conventional extracts of black cumin. All the results of the detected polyphenols are expressed as mean concentration (mg/g of dry extract) ± standard deviation (Table 4). According to our findings, six polyphenols were present in the UAE sample, but only four were above the limit of quantification. The concentrations of catechin, epicatechin, caffeic acid, rutin in the UAE sample were 0.18, 1.88, 0.17 and 0.96 mg/g, respectively. On the other hand, in the case of conventional extract, quantification was possible for five compounds, while gallic acid was below LOQ as well. In addition, the catechin, epicatechin, rutin contents were 0.27, 2.37 and 1.21 mg/g, respectively. However, the caffeic acid concentration in conventional extract exhibited the same level as the UAE sample (0.17 mg/g) as indicated in Figure 4.

Contrary to a previously published study [42] about the characterization of extracted phenolic acids and flavonoids from black cumin, quantification was possible for six compounds. Protocatechuic acid, caffeic acid, ellagic acid, ferulic acid, quercetin, kaempferol, and gallic acid were all measured at 0.13, 0.50, 0.15, 0.37, and 0.15 mg/g, respectively. In another study, by Saleha. H et al. [43], the HPLC analysis of black cumin seeds revealed the presence of chlorogenic acid, caffeic acid, kaempferol, and thymoquinone in concentrations of 5.5, 4.09, 6.02 and 5.13 μg/g, respectively. Additionally, our proof that substances such as rutin, gallic acid, and caffeic acid occur in black seeds also aligns with findings from a survey by Muzolf et al. [44]. In addition, a more recent study conducted in 2020 by Feng et al. [45] with the use of an LC-ESI-QTOF/MS system identified and quantified seven bioactive compounds, including kaempferol-3-glucoside, diosmin, quercetin, kaempferol, protocatechuic acid, p-hydroxybenzoic acid, and chlorogenic acid, at concentrations of 0.39, 0.21, 1.83, 9.8, 1.39, 22.86 and 0.02 mg/g, respectively. Previous studies indicated that gallic acid, caffeic acid, quercetin, and kaempferol are the most prevalent compounds in black cumin seeds. Based on our results, caffeic acid concentration was 0.17 mg/g in both dry extracts. Gallic acid was presented in both extracts at very low concentrations, below the limit of quantification, while quercetin was not found in any of them and kaempferol was not detected. Epicatechin was the major phenolic compounds in both extracts.

### 3.5. Optimized Condition and Comparison with Conventional Technique

Based on the experimental and predicted data, the optimum UAE conditions were of ethanol concentration of 59.1%, extraction temperature of 44.6 °C and extraction time of 32.5 min. Under these conditions, the TPC and DPPH radical scavenging activities were 35.6 (mg GAE/g) and 70.5%, respectively. In comparison with the conventional method, the TPC and DPPH radical scavenging activities in the optimal condition of UAE increased by about 42 and 21%, respectively (Table 5). The result revealed that the use of UAE has a beneficial effect on the extraction of bioactive compounds from black cumin seed, since it is greater than that obtained by conventional techniques. These results are primarily explained by the ability of ultrasonic extraction to promote mass transfer and speed up the extraction procedure, enhancing the extraction of bioactive compounds [21]. Additionally, the UAE was a quicker, more productive, and lower-temperature extraction process for recovering phenolics from defatted black cumin seeds.

## 4. Conclusions

The findings of the current study make it abundantly evident that extraction time, temperature, and solvent concentration have a significant impact on the contents of phenolic compounds present in black cumin defatted extracts, as well as their reported antioxidant activity. The highest amount of TPC and DPPH radical scavenging activities were 35.6 mg GAE/g and 70.5%, respectively. Moreover, the UAE optimization process demonstrates a strong influence of the sonication temperature, time and ethanol concentration with optimum conditions at 44.68 °C, 32.51 min and 59.18%, respectively. Additionally, according to the HPLC-UV phenolic profile, six polyphenols were found in the conventional extract, although only four of them were above LOQ, namely catechin (0.18 mg/g), epicatechin (1.88 mg/g), caffeic acid (0.17 mg/g), and rutin (0.96 mg/g). The extract obtained by UAE at the optimum conditions had greater total phenolic content and DPPH radical scavenging activity than that extracted by conventional technique. Overall, it was established that certain polyphenols included in this matrix had significant antioxidant capacity in accordance with the screening and characterization of the polyphenolic components that were found. As a result, it is assumed that black cumin seeds can be a potential source of bioactive chemicals and phenolic components that contribute to significant antioxidant activity.

## Figures and Tables

**Figure 1 biomolecules-12-01311-f001:**
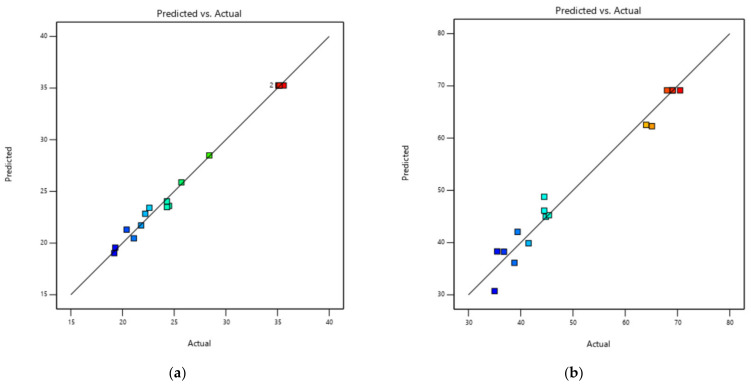
Diagnostic plots of predicted and actual values for TPC (**a**) and DPPH radical scavenging activity (**b**).

**Figure 2 biomolecules-12-01311-f002:**
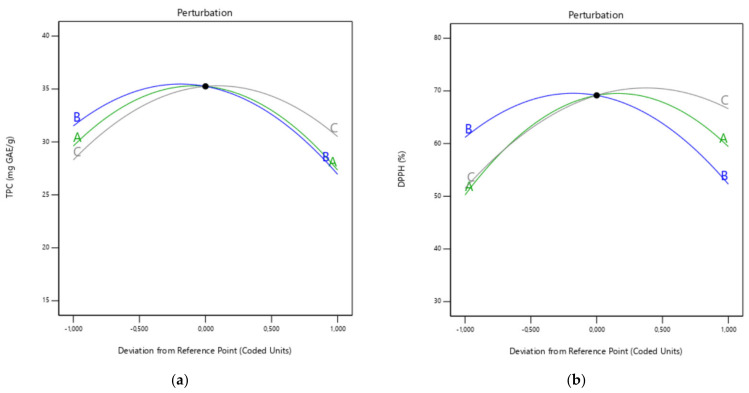
Perturbation plots showing the effect of extraction factors (A: ethanol concentration; B: temperature; C: extraction time) on TPC (**a**) and DPPH radical scavenging activity (**b**).

**Figure 3 biomolecules-12-01311-f003:**
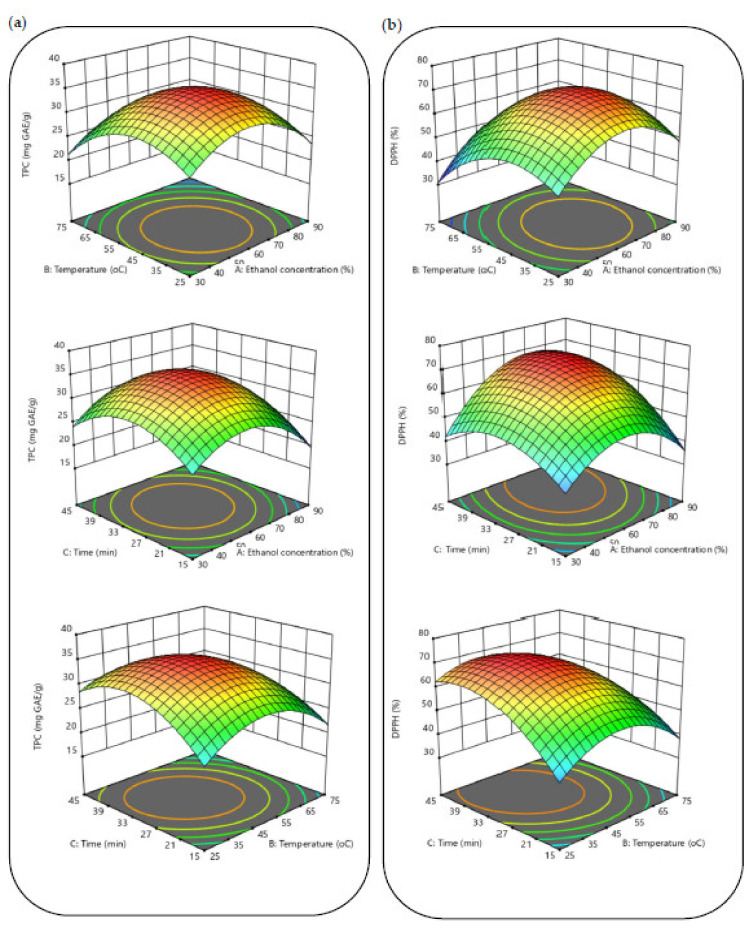
3D response surface plots showing the interactive effects of process factors on TPC (**a**) and DPPH (**b**).

**Figure 4 biomolecules-12-01311-f004:**
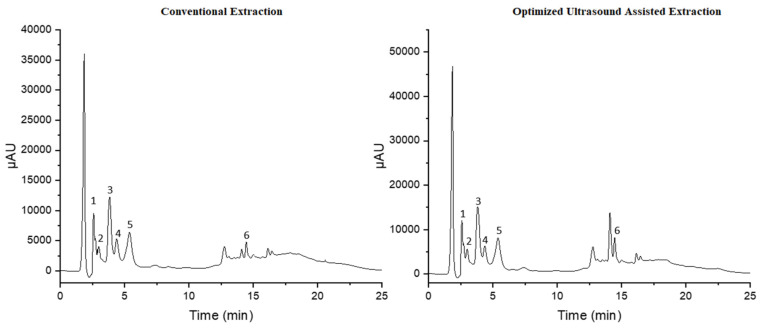
HPLC chromatograms of phenolic compounds for extracts obtained by conventional extraction and ultrasound-assisted extraction under the optimum conditions. (1) Gallic acid, (2) catechin, (3) epicatechin, (4) Caffeic Acid, (5) Rutin and (6) Quercetin.

**Table 1 biomolecules-12-01311-t001:** Box–Behnken design matrix with coded variables and measured values.

	Factors			Responses	
Run	X_1_: Ethanol Concentration (%)	X_2_: Temperature (°C)	X_3:_ Time (min)	Y_1_: TPC (mg GAE/g)	Y_2_: DPPH (%)
1	60	25	45	28.4	65.1
2	30	25	30	25.7	44.8
3	60	75	15	21.8	35.5
4	60	50	30	35.1	70.5
5	30	50	45	24.3	39.4
6	90	25	30	24.5	44.5
7	30	50	15	24.3	36.8
8	60	50	30	35.6	68
9	60	75	45	21.1	44.5
10	60	50	30	35.2	69.1
11	60	50	30	35.1	69
12	60	25	15	22.2	41.5
13	90	75	30	19.2	45.4
14	90	50	45	22.6	64
15	30	75	30	20.4	35
16	90	50	15	19.3	38.8

**Table 2 biomolecules-12-01311-t002:** Analysis of variance for the regression model of TPC and DPPH.

	TPC			DPPH		
Source	Coefficient Estimate	F-Value	*p*-Value	Coefficient Estimate	F-Value	*p*-Value
Model	35.25	87.37	<0.0001	69.15	23.84	0.0005
X_1_-Ethanol con.	−1.14	15.01	0.0082	4.59	12.75	0.0118
X_2_-Temperature	−2.29	60.71	0.0002	−4.44	11.93	0.0136
X_3_-Time	1.10	14.04	0.0095	7.55	34.53	0.0011
X_1_X_2_	0.0000	0.0000	1.0000	2.67	2.17	0.1914
X_1_X_3_	0.8250	3.95	0.0941	5.65	9.67	0.0209
X_2_X_3_	−1.72	17.26	0.0060	−3.65	4.03	0.0913
X_1_^2^	−6.78	266.25	<0.0001	−14.31	62.04	0.0002
X_2_^2^	−6.02	210.57	<0.0001	−12.41	46.66	0.0005
X_3_^2^	−5.85	198.51	<0.0001	−10.09	30.82	0.0014
Lack of Fit	-	23.34	0.0140	-	24.00	0.0134
R^2^	0.9924			0.9728		
Adjusted R^2^	0.9811			0.9320		
C.V.%	3.20			7.16		
Adeq Precision	24.71			13.37		

**Table 3 biomolecules-12-01311-t003:** Limits of detection and quantification, regression equations, retention times, repeatability and reproducibility (*n* = 3).

Compounds	Rt ± SD	Linear Range	Linear Equation	R^2^	LOD (μg/mL)	LOQ (μg/mL)	Intraday RSD_Area_%	Interday RSD_Area_%
Gallic Acid	2.52 ± 0.00	0.01–0.0001	y = 84073550x + 10609	0.998	0.3	0.9	1.00	0.92
Catechin	3.34 ± 0.02	0.05–0.0001	y = 13269933x − 2012	0.999	0.3	0.9	0.95	2.53
Epicatechin	3.83 ± 0.03	0.01–0.0001	y = 16600346x + 2404	0.998	0.3	0.8	1.04	1.47
Caffeic Acid	4.54 ± 0.05	0.01–0.0001	y = 55118151x + 8713	0.997	0.3	1.0	0.47	196
Rutin	5.54 ± 0.07	0.05–0.0001	y = 23176073x − 12698	0.998	0.9	2.8	0.55	1.80
Sinapic Acid	7.99 ± 0.06	0.01–0.0001	y = 39382578x + 2882	0.997	0.4	1.1	0.33	1.92
Quercitrin	11.49 ± 0.08	0.01–0.0001	y = 12295377x − 1169	0.999	0.1	0.3	0.42	1.84
Myricetin	13.12 ± 0.03	0.01–0.0001	y = 25427092x + 2679	0.998	0.3	0.9	0.90	2.68
Quercetin	14.67 ± 0.04	0.01–0.0001	y = 34063703x + 42715	0.998	0.3	0.9	0.32	1.67

**Table 4 biomolecules-12-01311-t004:** Concentration of the detected phenolics in black cumin expressed as mg/g of dry extract (*n* = 3).

Phenolic Compound	Conventional Technique (mg/g)	UAE Technique at Optimum Conditions (mg/g)
Gallic Acid	NQ	NQ
Catechin	0.18 ± 0.00	0.27 ± 0.00
Epicatechin	1.88 ± 0.00	2.37 ± 0.02
Caffeic Acid	0.17 ± 0.00	0.17 ± 0.00
Rutin	0.96 ± 0.00	1.21 ± 0.01
Sinapic Acid	ND	ND
Quercitrin	ND	ND
Myricetin	ND	ND
Quercetin	NQ	0.15 ± 0.00

ND: not detected. NQ: not quantified.

**Table 5 biomolecules-12-01311-t005:** Comparison in extraction conditions, TPC and DPPH of each technique.

Methods	Ethanol Concentration (%)	Temperature (°C)	Time (min)	TPC (mg GAE/g)	DPPH (%)	IC50 Values for DPPH (mg/mL)
UAE-optimized	59.1	44.6	32.5	35.6	70.5	1.14
Conventional	50	50	120	20.9	51.1	1.96

## Data Availability

The data presented in this study are available on request from the corresponding authors.

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
