# Peer review of "Phenolics from Defatted Black Cumin Seeds (Nigella sativa L.): Ultrasound-Assisted Extraction Optimization, Comparison, and Antioxidant Activity"

_biomolecules, 2022, doi:10.3390/biom12091311_

Round 1
Reviewer 1 Report
Interesting research. Congrats.
All my comments and suggestions are included in attached pdf file.
Kind regards.

Author Response
We warmly appreciate all your valuable comments and suggestions, which helped us to improve the quality of the manuscript. we have carefully corrected the manuscript according to each comment. Your requested corrections carried out in this second draft are in track changes.
Reviewer 2 Report
Dear authors,
thank you for this interesting paper. I have several questions to you:
1. Why do you express antiradical activity in vitro as percentage of DPPH• scavening? In my personal opinion it is not fully correct. Also it is quite difficult to compare your data to other scientists. Please express antiradical activity as equivalent of standart antioxidant (for example Trolox, vitamin C or other).
2. Why do you calculate LOD and LOQ using formulas 3.3 σ/ Slope and LOQ= 10 σ/ Slope? I think it could be better calculate LOD and LOQ using signal to noise from your chromatogram. In addition to this why you do not write any references where you found the formulas? I think the references must be added here (for example ICH guidelines or other references).
3. Where is the chromatogram of the separation of the phenolic compounds? I did not find it in the paper. You must add the chromatogram.
4. Why did you choose sonification and convention extraction? Why you did not compare more extraction methods (for example microwave extraction, percolation and etc.)? Could you expand your experiment?
Author Response
Thank you for this interesting paper. I have several questions to you:
We would like to thank you for the time and effort taken to review our manuscript, which helped us to improve the quality of the manuscript. Your requested corrections carried out in this second draft are in track changes.
- Why do you express antiradical activity in vitroas percentage of DPPH• scavening? In my personal opinion it is not fully correct. Also it is quite difficult to compare your data to other scientists. Please express antiradical activity as equivalent of standart antioxidant (for example Trolox, vitamin C or other).
DPPH radical scavenging activity expressed as percentage for optimization by using box-Behnken design to demonstrate the effect of extraction conditions on antioxidant activity more clearly. According to your suggestion, the IC-50 value for DPPH of the extracts has been added to the table 5 to make it easier to compare with other studies.
- Why do you calculate LOD and LOQ using formulas 3.3 σ/ Slope and LOQ= 10 σ/ Slope? I think it could be better calculate LOD and LOQ using signal to noise from your chromatogram. In addition to this why you do not write any references where you found the formulas? I think the references must be added here (for example ICH guidelines or other references).
The particular formulae of LOD and LOQ are widely used in highly rated internationally recognized analytical journals such as follows:
https://doi.org/10.1016/j.chroma.2021.462035
https://doi.org/10.3390/molecules26165017
The intended reference has been mentioned.
- Where is the chromatogram of the separation of the phenolic compounds? I did not find it in the paper. You must add the chromatogram.
HPLC chromatograms have been added to the manuscript (Figure 4)
- Why did you choose sonification and convention extraction? Why you did not compare more extraction methods (for example microwave extraction, percolation and etc.)? Could you expand your experiment?
The aim of this project was to optimize ultrasound-assisted extraction as an efficient green technique for the extraction of phenolic compounds from black cumin seeds. In another part of this project next year, we are going to optimize the microwave-assisted extraction for several medicinal plants. However, due to the thermo-sensitivity of some phenolic compounds, ultrasonic technology could be more efficient for extraction of phenolic compound-rich extracts compared to other techniques.
We apologize for missing this important point in this article. Due to the fact that the tested samples are not available right now, it is not possible to perform the test. In future studies, we will definitely make this valuable point in our articles.